# Revealing the Real Role of Etching during Controlled Assembly of Nanocrystals Applied to Electrochemical Reduction of CO_2_

**DOI:** 10.3390/nano12152546

**Published:** 2022-07-24

**Authors:** Tingting Yue, Ying Chang, Haitao Huang, Jingchun Jia, Meilin Jia

**Affiliations:** College of Chemistry and Environmental Science, Inner Mongolia Key Laboratory of Green Catalysis and Inner Mongolia Collaborative Innovation Center for Water Environment Safety, Inner Mongolia Normal University, Hohhot 010022, China; yuett9707@163.com (T.Y.); huanght2022@163.com (H.H.); jml@imnu.edu.cn (M.J.)

**Keywords:** N-Fe_3_C/rGO-H, electrochemical CO_2_ reduction, synergistic effect

## Abstract

In recent years, the use of inexpensive and efficient catalysts for the electrocatalytic CO_2_ reduction reaction (CO_2_RR) to regulate syngas ratios has become a hot research topic. Here, a series of nitrogen-doped iron carbide catalysts loaded onto reduced graphene oxide (N-Fe_3_C/rGO-H) were prepared by pyrolysis of iron oleate, etching, and nitrogen-doped carbonization. The main products of the N-Fe_3_C/rGO-H electrocatalytic reduction of CO_2_ are CO and H_2_, when tested in a 0.5 M KHCO_3_ electrolyte at room temperature and pressure. In the prepared catalysts, the high selectivity (the Faraday efficiency of CO was 40.8%, at −0.3 V), and the total current density reaches ~29.1 mA/cm^2^ at −1.0 V as demonstrated when the mass ratio of Fe_3_O_4_ NPs to rGO was equal to 100, the nitrogen doping temperature was 800 °C and the ratio of syngas during the reduction process was controlled by the applied potential (−0.2~−1.0 V) in the range of 1 to 20. This study provides an opportunity to develop nonprecious metals for the electrocatalytic CO_2_ reduction reaction preparation of synthesis and gas provides a good reference

## 1. Introduction

The Earth’s temperature is rising, and extreme weather, forest fires and ice shelf collapses are a growing problem. Curbing climate warming has become an urgent issue for the international community. CO_2_, as the most dominant greenhouse gas, is considered to be the breakthrough point to curb warming. The concept of “carbon neutrality” was born. Among the many carbon neutral technology pathways, there is one governance idea that has attracted particular attention due to its high efficiency in reducing CO_2_ concentrations in the atmosphere, namely, carbon capture and resource utilization. CO_2_ is an extremely stable molecule that is involved in chemical synthesis as a raw material and requires the absorption of large amounts of energy [1]. Therefore, transforming CO_2_ into syngas under mild conditions is an urgent need for protecting the environment and reducing energy consumption. [2]. The electrocatalytic CO_2_ reduction reaction (CO_2_RR) is mild, does not require high temperature and pressure and only requires electricity consumed from sustainable energy sources to convert CO_2_ to CO [3,4]. In addition, some electricity waste is inevitable due to the cyclical nature of solar and wind power [5]. This process is economically feasible through the use of excess CO_2_ and waste electricity [6,7,8].

CO_2_RR in aqueous solution is always combined with hydrogen evolution reaction (HER); therefore, H^+^ in water is used as a source of hydrogen to prepare syngas [9,10,11]. Syngas is a mixture of hydrogen (H_2_) and carbon monoxide (CO) that can be used as a raw material or intermediate in chemical synthesis, such as Fischer–Tropsch synthesis for the preparation of alcohols and olefins. In recent years, researchers have designed a large number of catalysts for the preparation of syngas from CO_2_RR [12,13]. Precious metals, such as Au [14], Ag [15] and Pd_3_Bi [16], and other metals, such as Zn [17], Pb/CNT [18], Co_3_O_4_-CDots-C_3_N_4_ [19], CdS_x_Se_1−x_ [20], E-MoS_2_ [21] and Cu-Sn alloys [22], have been reported, with the latter making great progress. For example, Zou et al. [23] reported Ni-Co double monoatomic loading to carbon nanotubes for the preparation of syngas with CO/H_2_ ratios between 1.3 and 1.5 and Ni-N_4_ and Co-N_4_ as the active sites for the CO_2_RR and HER, respectively. Wang’s group [8] converted CO_2_ to syngas by selective phosphorylation of reduced partial Cu_3_P nanowires, and the vacancies created by phosphorylation accelerated charge transfer and significantly improved the activity and selectivity of CO_2_. However, achieving high current density applications in aqueous solutions is still difficult in practice and limited in scale due to the small reserves of precious metals in the Earth’s crust. Therefore, the search for catalysts that are inexpensive and suitable for application in CO conversion to H_2_/CO in Fischer–Tropsch synthesis is urgent.

Among many nonprecious metals, carbon-based materials intrigue researchers in CO_2_RR due to their cheap price, large specific surface area and high electrical conductivity [24,25]. In particular, transition metals loaded onto N-doped carbon substrates (M-N-C) have attracted the attention of researchers as new materials. Zhao et al. [26] prepared a series of Fe-N-C catalysts to control the ratio of H_2_/CO in the range of 4:1 to 1:3. Yue et al. [27] designed the in situ loading of Fe-Ni alloys onto N-doped carbon substrates by changing the applied potential and adjusting the Fe-Ni molar ratio to flexibly modulate the syngas ratio. Jia’s group [28] developed a N-Fe_3_C/rGO catalyst for CO_2_RR syngas preparation and found that the synergistic effect of C-N and Fe-N facilitated CO generation, while rGO as a carrier improved the electrical conductivity. To date, M-N-C catalysts for oxygen reduction reactions have also shown efficient catalytic activity [29,30]. Although M-N-C for CO_2_RR can produce CO, HCOOH and C_2_H_4_ [31,32,33], there are relatively few studies based on low-cost M-N-C materials to investigate the ratio of syngas fractions through adjustment of the applied electric potential. Here, we prepared Fe_3_O_4_ NPs using iron oleate precursors and assembled them into rGO with different mass ratios and then doped with nitrogen at different temperatures to investigate the effects of Fe_3_O_4_/rGO and nitrogen doping temperature on the CO_2_RR. The catalysts prepared at mass ratio of 100 and 800 °C were applied to the CO_2_RR, showing good activity and selectivity.

## 2. Results and Discussion

The iron oleate precursors were first prepared using FeCl_3_·6H_2_O and NaOA oil baths, and the precursors were pyrolyzed to generate Fe_3_O_4_ NPs [34], which would be assembled into rGO with different mass ratios of nanoparticles, as shown in Appendix A. The SEM images of Fe_3_O_4_/rGO obtained after using HCl etching to obtain Fe_3_O_4_/rGO-H are shown in Appendix A. The successful loading of Fe_3_O_4_ onto the rGO surface can be observed from the SEM image. Comparing the distribution of Fe_3_O_4_ NPs on the rGO surface before and after etching, it was found that the NPs discretely stacked and then agglomerated with increasing nanoparticle loading before etching; after strong acid etching, the number of nanoparticles decreased significantly, and a hole-like structure appeared. The addition of a nitrogen source continued the sintering at high temperature, and it was found that the dispersion of nanoparticles was better at 800 °C with an Fe_3_O_4_/rGO mass ratio of 100, and agglomeration started to occur at 120, as illustrated Figure 1. Comparison of the SEM images for different nitrogen doping temperatures showed no influence of temperature on the morphology of the catalyst.

Figure 2a,b show the XRD patterns of Fe_3_O_4_/rGO before and after etching with strong acid. The diffraction peaks are located at 30.0°, 35.4°, 43.1°, 56.9° and 62.5° according to the (220), (311), (400), (511) and (440) Fe_3_O_4_ crystal planes (PDF#99-0073), respectively. It was observed that the crystal planes did not change before and after etching, but the peak intensity was slightly weakened. XRD patterns of N-Fe_3_C/rGO-H obtained by calcination of Fe_3_O_4_/rGO doped with dicyandiamine at high temperature are shown in Figure 2c. The 2θ positions of the peaks are at 26.4°, 37.6°, 39.8°, 40.6°, 42.8°, 43.7°, 44.5°, 44.9°, 45.8°, 49.1°, 51.8°, 54.4°, 58.4° and 70.8° corresponding to Fe_3_C’s (020), (121), (002), (201), (211) (102), (220), (031), (112), (221), (122), (230), (231) and (123) sides of Fe_3_C (PDF#35-1-0772), respectively. The XRD results demonstrate the successful preparation of Fe_3_O_4_ NPs by pyrolysis of iron oleate precursors, which were converted to Fe_3_C by sintering at high temperature after nitrogen doping. The XRD of the comparison samples N-Fe_3_C/rGO-H-700 and N-Fe_3_C/rGO-H-900 are shown in Appendix A, and the calcination temperature has no effect on the formation of the crystalline phase of Fe_3_C.

TEM further observed the morphological changes of Fe_3_O_4_/rGO before and after strong acid etching and nitrogen-doped high-temperature calcination. Figure 3a shows the integration of nanoparticles on the carrier surface after calcination of Fe_3_O_4_ NPs and rGO components at 500 °C. Since the number of nanoparticles anchored on the carrier surface was limited, the excess and poorly anchored particles on the rGO surface were etched off with 1 M HCl, as shown in Figure 3d. Observation of the pictures revealed that the remaining Fe_3_O_4_ NPs were more uniformly dispersed on the carrier surface after the accumulation of particles in rGO was successfully etched off. Then, dicyandiamide was added and scorched at 800 °C. The Fe atoms overflowed on the rGO surface, leaving a hollow shell (Figure 3g). These shell structures may become the active sites of the CO_2_RR. In addition, the Fe_3_O_4_/rGO-100 and Fe_3_O_4_/rGO-H-100 grain sizes are approximately 10.2 ± 1.2 nm and 9.73 ± 1.2 nm, respectively, and the grain size of the empty shells is approximately 6.74 ± 1.2 nm. The grain size data indicate that the decrease in the number of iron atoms on the substrate and the high-temperature carbonization cause the nanoparticle size and the size of the empty shells to gradually decrease. By measuring the lattice stripes in the HRTEM of Fe_3_O_4_/rGO-100, Fe_3_O_4_/rGO-H-100 and N-Fe_3_C/rGO-H-100 as 0.241 nm, 0.200 nm and 0.288 nm, 0.195 nm responding to the (222), (400), (220) planes of Fe_3_O_4_ and (112) faces of Fe_3_C, respectively. The diffraction rings of Figure 3c,f,i illustrate that all three nanoparticles are polycrystalline in structure. The energy-dispersive X-ray spectroscopy (EDS) of N-Fe_3_C/rGO-H-100 clearly shows that C, Fe and N are uniformly distributed on the rGO surface (Figure 3j–m).

The interaction between rGO and Fe_3_C was studied and the Raman spectra of the catalysts were determined (Appendix A). One band at approximately 1350 cm^−1^ (D-band) corresponds to the hexagonal graphene plane in the samples tested. Another band at about 1586 and 1580 cm^−1^ (G-band) is attributed to defects in N-Fe_3_C/rGO-H-100, and rGO, respectively [35]. I_D_/I_G_ (the ratio of D-band to G-band intensity) is used to measure the disorder of carbon materials [36]. The I_D_/I_G_ value of N-Fe_3_C/rGO-H-100 (1.03) is greater than that of rGO, suggesting that the loading of Fe_3_C onto the carbon-based material leads to an increase in disorder leading to an increase in defects and the acquisition of more active sites thus improving the performance of CO_2_RR [26].

Using XPS to understand the composition and valence of the catalyst, it can be seen from Figure 4a that it mainly contains four elements, C, N, Fe and O, without other spurious peaks, in agreement with the mapping results. As shown in Figure 4b, the fine spectra of C1s coincide with peaks positioned at 284.6, 285.7 and 286.6 eV, corresponding to C=C, C-N and C=O, respectively [37]. The characteristic peaks of the high-resolution spectra of N1s were located at 398.5, 399.7, 400.8 and 403.9 eV, corresponding to pyridine N, Fe-N, graphite N and oxidized N, respectively, as shown in Figure 4c [38,39]. The Fe-N peak is observed at 399.7 eV, indicating that an iron atom is coupled with two N atoms [40]. Figure 4d exhibits the Fe 2P spectrum of N-Fe_3_C/rGO-H-100. The binding energies at 710.3 and 723.2 eV would correspond to 2P_3/2_ and 2P_1/2_ of Fe^2+^; those located at 713.4 and 725.4 eV would correspond to 2P_3/2_ and 2P_1/2_ of Fe^3+^ [41]. Furthermore, 710.3 eV is also considered the chemical shift of Fe coordinated to N, which is consistent with the analysis of N1s. Fe-N_X_ provides N doping that provides strong evidence and can enhance the charge transfer between rGO and Fe [30,42,43]. The peak at 707.8 eV can be considered the peak of Fe^0^ or iron carbide, and the weaker peak intensity is due to the low density of metallic iron agglomerates and the encapsulation of the FeN_x_ active site by the carbon substrate [44,45,46].

Both the mass ratio of Fe_3_O_4_/rGO and the nitrogen-doped carbonization temperature affect the performance of the CO_2_RR. As shown in Figure 5, all electrochemical tests were performed in a 0.5 M KHCO_3_ solution saturated with CO_2_. Three tests were performed at the same time interval at each applied point and then averaged. The prepared catalyst N-Fe_3_C/rGO-H produces CO and H_2_ as the main products for the reduction of CO_2_. The Faraday efficiency at different mass ratios is closely related to the applied potential. Figure 5a depicts the effect of Fe_3_O_4_/rGO at different mass ratios. It can be observed that the FE_CO_ of N-Fe_3_C/rGO-H-100 gradually increases, reaching a maximum value of 40.8% when the potential is from −0.2 to −0.3 V. While at −0.4 to −1.0 V, the FE_CO_ gradually decreases as the potential becomes negative. The CO_2_RR and HER are competing reactions; therefore, FE_H2_ shows the opposite trend. In Figure 5b, FE_H2_ first decreases sharply to 37.0% at −0.2 to −0.3 V and then progressively increases with increasing potential. The maximum FE potential for the reduction of CO_2_ to CO by N-Fe_3_C-H was located at −0.5 V, in contrast to the lower catalytic activity and selectivity of rGO for the CO_2_RR, indicating that N-Fe_3_C/rGO-H can provide the catalytic active site to promote the CO_2_RR. Compared to N-Fe_3_C/rGO-H (mass ratios of 80 and 120), N-Fe_3_C-H and rGO, N-Fe_3_C/rGO-H-100 exhibited good activity for the CO_2_RR. This is because small loadings do not provide enough catalytic active sites, and too high loadings cause the active sites to accumulate [47].

Therefore, N-Fe_3_C-H, rGO, and N-Fe_3_C/rGO-H-100 were performed by linear scanning voltammetry (LSV) at a sweep rate of 50 mV/s (Figure 5c). N-Fe_3_C/rGO-H-100 exhibited positive onset and high current density over the entire potential range tested, indicating its high catalytic activity in the CO_2_RR [48,49]; the current density of each catalyst at different potentials was also tested, as shown in Appendix A, and it was found that rGO increased the current density of N-Fe_3_C; this also implies that the synergistic results of N-Fe_3_C and rGO can improve the catalytic performance. Combined with XPS analysis, it can be speculated that C-N and Fe-N are the catalytic active sites of the CO_2_RR. This is because the introduction of N atoms into the carbon substrate provides an optional opportunity to directly tune the electronic structure of the metal centre of the anchored molecular catalyst. The doping of N in the conducting carbon substrate as an axial ligand for the iron centre reduces the electron density of the iron 3D orbitals, thus reducing the Fe-CO π back-donation, facilitating the desorption of CO and improving the electrocatalytic activity and selectivity [50,51,52]. The bilayer capacitance (Cdl) of the three catalysts was tested using cyclic voltammetry; since the bilayer capacitance is proportional to the electrochemically active specific surface area (ECSA), Cdl can be used to represent the ECSA. The interval without any Faraday potential is selected and tested with different sweeps, as shown in Appendix A. The Cdl values of N-Fe_3_C-H, rGO and N-Fe_3_C/rGO-H-100 were 5.4, 3.9 and 6.5 mF/cm^2^, respectively. The Cdl of N-Fe_3_C/rGO-H-100 was significantly better than that of N-Fe_3_C-H and rGO, so its high catalytic activity contributed to having a large ECSA. We investigated the stability of N-Fe_3_C/rGO-H-100 for CO_2_ reduction at −0.4 V (Appendix A). The FE_CO_ was found to remain at approximately 26% after 12 h, indicating that the catalyst still has high catalytic activity and its morphology remains largely unchanged after a long stability test. As shown in Appendix A, by varying the applied potential (−0.2 to −1.0), the syngas H_2_/CO ratio varies from 1 to 20 and was compared with other recently reported catalysts for syngas preparation (Appendix A). It can be clearly shown that the desired syngas ratio can be obtained by adjusting the potential.

We found a suitable Fe_3_O_4_/rGO mass ratio (100), and, at this mass ratio, the effect of nitrogen doping temperature on the CO_2_RR performance was investigated. Appendix A shows the LSV of N-Fe_3_C/rGO-H at different nitrogen doping temperatures. The onset potentials are similar for temperatures of 700 and 800 °C, but the current density of N-Fe_3_C/rGO-H-100 is slightly higher than that of N-Fe_3_C/rGO-H-700, as further demonstrated in Appendix A. To further investigate the effect of temperature on the CO_2_RR, electrochemical tests were performed using these three catalysts and compared (Appendix A). The FE_CO_ of N-Fe_3_C/rGO-H-100 increases and then decreases as the potential becomes negative, reaching a maximum of 40.8% at −0.3 V. Similarly, the corresponding FE_H2_ decreases to a minimum value of 37.0%. This is because different nitrogen doping temperatures affect the active sites formed between Fe, N and C, and, hence, show variable catalytic properties. The level of nitrogen doping temperature affects the active site of the reaction, and only the right temperature can show the best catalytic performance.

Electrochemical tests reveal that: the assembly of different masses of Fe_3_O_4_ NPs to rGO for CO_2_RR results in a consistent trend of CO generation (FE reaches a maximum at −0.3 V) as shown in Appendix A. The applied potential of the maximum FE corresponding to the reduction of CO_2_ to CO by N-Fe_3_C-H increases (−0.5 V), indicating a lower activity than N-Fe_3_C/rGO-H. Pure rGO is dominated by the hydrogen evolution reaction.

Based on the above expression, the optimal nitrogen-doped carbonization temperature and mass ratio are 800 °C and 100, with 40.8% and 37.0% FE_CO_ and FE_H2_ for catalyst N-Fe_3_C/rGO-H-100 at −0.3 V, respectively (Figure 5d). The reasons for the high performance of N-Fe_3_C/rGO-H-100 on the CO_2_RR are as follows: (1) the high catalytic activity after strong acid etching of part of Fe_3_C leaving empty shells on the rGO surface and the interaction between Fe, N and C to regulate the electron and mass transport [26]; (2) the bonding of the metal to the nitrogen site causes an increase in the density of positive charges on the adjacent carbon atoms, which facilitates the doping of the metal Fe-N-C active site [53]; and (3) rGO acts as a catalyst substrate to provide good reaction conditions and improve the conductivity of the reaction.

## 3. Conclusions

In this study, N-Fe_3_C/rGO-H-100 was used for the electrocatalytic CO_2_ reduction reaction to regulate the syngas ratio. The electrochemical test results showed that the reduction of CO_2_ exhibited good activity at lower potentials. At −0.3 V. The Faraday efficiency of CO was 40.8%, and the syngas ratio was adjusted to 1–20 by adjusting the applied potential (−0.2~−1.0 V); and the total current density reached 29.1 mA/cm^2^ at −1.0 V. N-Fe_3_C/rGO-H-100 has good activity, selectivity and long-term stability at relatively low potentials and can be used as an effective catalyst for the electrocatalytic CO_2_ reduction reaction.

## Figures and Tables

**Figure 1 nanomaterials-12-02546-f001:**
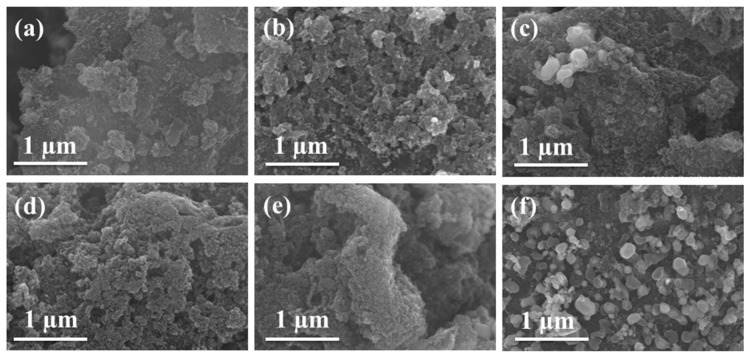
SEM images of (**a**) N-Fe_3_C/rGO-H-80, (**b**) N-Fe_3_C/rGO-H-100, (**c**) N-Fe_3_C/rGO-H-120, (**d**) N-Fe_3_C/rGO-H-700, (**e**) N-Fe_3_C/rGO-H-900, and (**f**) N-Fe_3_C-H.

**Figure 2 nanomaterials-12-02546-f002:**
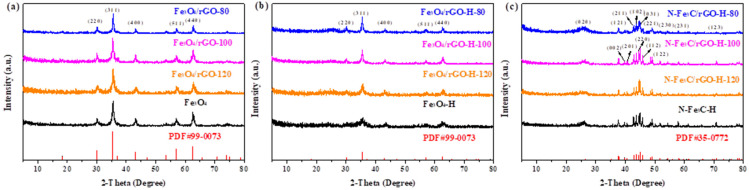
XRD patterns of (**a**) Fe_3_O_4_/rGO, (**b**) Fe_3_O_4_/rGO-H and (**c**) N-Fe_3_C/rGO-H.

**Figure 3 nanomaterials-12-02546-f003:**
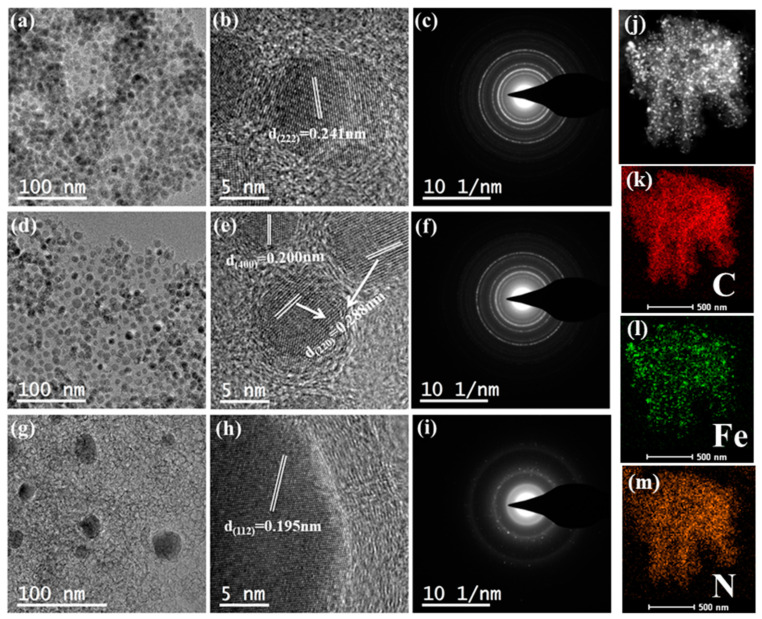
TEM images of (**a**–**c**) Fe_3_O_4_/rGO-100, (**d**–**f**) Fe_3_O_4_/rGO-H-100 and (**g**–**i**) N-Fe_3_C/rGO-H-100. (**j**–**m**) EDS mapping of C, Fe and N in N-Fe_3_C/rGO-H-100.

**Figure 4 nanomaterials-12-02546-f004:**
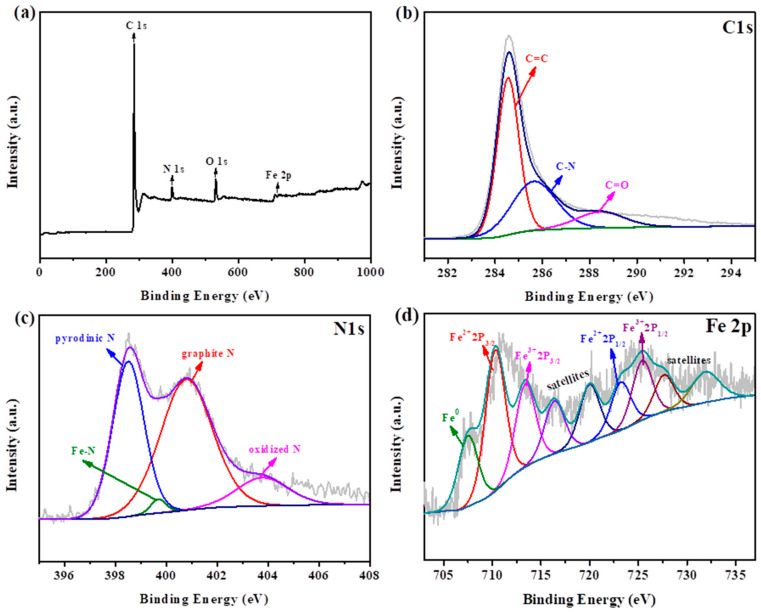
XPS spectra of N-Fe_3_C/rGO-H-100: (**a**) survey spectra, (**b**) C 1s, (**c**) N 1s and (**d**) Fe 2p.

**Figure 5 nanomaterials-12-02546-f005:**
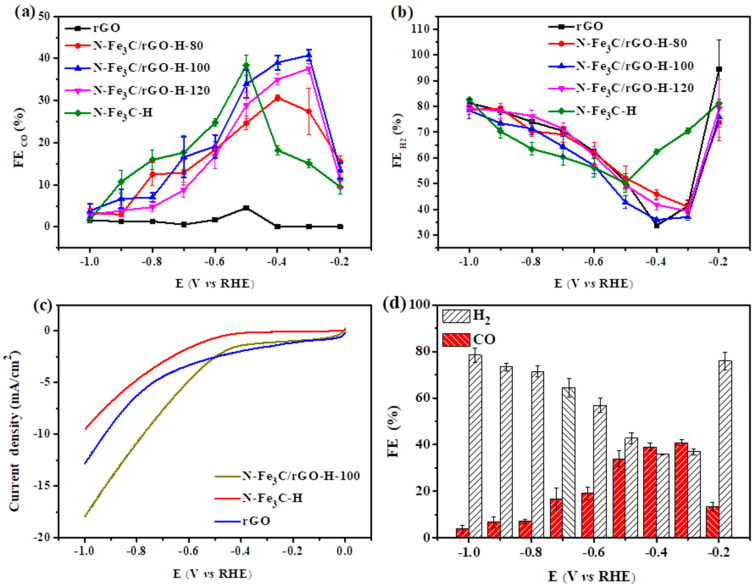
(**a**,**b**) Faradaic efficiency of CO and H_2_ at different mass ratios; (**c**) LSV of N-Fe_3_C/rGO-H-100, N-Fe_3_C-H and rGO; (**d**) FE of N-Fe_3_C/rGO-H-100.

## Data Availability

Not applicable.

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
