# Peer review of "Revealing the Real Role of Etching during Controlled Assembly of Nanocrystals Applied to Electrochemical Reduction of CO2"

_nanomaterials, 2022, doi:10.3390/nano12152546_

Round 1

Reviewer 1 Report

The paper describes synthesis and characterization of nitrogen-doped iron carbide catalysts loaded  onto reduced graphene oxide for CO2 electrochemical reduction. The paper requires major modifications as many essential details are missing.

General comments:

1) The abstract should be made more informative (including catalyst type, electrochemical testing conditions, electrolyte, measured reaction rate and other important details)

2) Nothing is mentioned about the details of the electrochemical cell (distance between the electrodes, the size of the Pt electrode and the position of the counter electrode). Please elaborate as this is important to estimate the Ohmic losses in the system which could contribute to overall losses when you discuss the current density (Figure 5). How the pressure was regulated in the cell as the gas was produced by electrochemical reaction?

Specific comments:

3) units are missing in the electrode size: 1*1 carbon paper (Shanghai Hesen Reagent).

4) Line 98.  A long list of (020), (121), (002), (201), (211) (102), 98 (220), (031), (112), (221), (122), (230), (231) and (123). This is difficult to follow, it is better to indicate those numbers directly in the Figure

5) Line 117. The accuracy of numbers providing the grain size needs to be reduced to one significant number after the comma and a standard deviation should be added.

6) Line 125 “ The TEM corresponding elemental distribution map” The TEM cannot produce distribution map, please reword this sentence.

7) Line 137. The discussion for Fe should mention its oxidation state (and supported by reference data). It is not clear what “oxidized N” is? Please elaborate in more details.

8) Line 154 “The Faraday efficiency (FE) at different mass ratios is closely related to the applied spot position” What is spot position? This needs to be explained.

9) Line 177 “it can be speculated that C-N and Fe-N are the catalytic active sites of the CO2RR”. It is not clear how this conclusion could be made. More discussion is required and what are the sorption mechanism onto these active sites?

10) Line 217 “that the reduction of CO2 exhibited good activity at lower potentials” Could you specify reaction rate explicitly.

11) SI.  "Put 10 mL of electrolyte in each cathode". This is not clear.

Author Response

  • The abstract should be made more informative (including catalyst type, electrochemical testing conditions, electrolyte, measured reaction rate and other important details)

Response: We thank the reviewer for her/his comments and suggestions. We have added more information to the abstract. (Please see line 10-15)

  • Nothing is mentioned about the details of the electrochemical cell (distance between the electrodes, the size of the Pt electrode and the position of the counter electrode). Please elaborate as this is important to estimate the Ohmic losses in the system which could contribute to overall losses when you discuss the current density (Figure 5). How the pressure was regulated in the cell as the gas was produced by electrochemical reaction?

Response: We thank the reviewer for her/his comments and suggestions. We have added details of electrochemical cells to the supporting information. (Please see SI lines 49-59). In this paper, no resistance compensation was carried out and the conductivity was improved by reducing the distance between the reference electrode and the working electrode in order to reduce ohmic consumption and by replacing the 0.1 M KHCO3 electrolyte with 0.5 M KHCO3.The electrolytic cell used in our tests has an inlet and outlet port at the top of the cathode chamber, which allows the pressure in the cell to be regulated as the gas is generated during the reaction and fed into the gas chromatograph along with the carrier gas (CO2).

Specific comments:

  • units are missing in the electrode size: 1*1 carbon paper (Shanghai Hesen Reagent).

Response: We thank the reviewer for her/his comments and suggestions. We should complete the carbon paper units in the supporting information. (Please see SI lines 50)

  • Line 98.  A long list of (020), (121), (002), (201), (211) (102), 98 (220), (031), (112), (221), (122), (230), (231) and (123). This is difficult to follow, it is better to indicate those numbers directly in the Figure2.

Response: We thank the reviewer for her/his comments and suggestions. We have marked the numbers in Figure 2 directly on the graph. (Please see Figure 2)

  • Line 117. The accuracy of numbers providing the grain size needs to be reduced to one significant number after the comma and a standard deviation should be added.

Response: We thank the reviewer for her/his comments and suggestions. We have reduced the need for accuracy in the numbers providing granularity to one valid number after a comma and added standard deviations. (Please see lines 118-120)

  • Line 125 “ The TEM corresponding elemental distribution map” The TEM cannot produce distribution map, please reword this sentence.

Response: We thank the reviewer for her/his comments and suggestions. We have changed this sentence. (Please see lines 126-127)

  • Line 137. The discussion for Fe should mention its oxidation state (and supported by reference data). It is not clear what “oxidized N” is? Please elaborate in more details.

Response: We thank the reviewer for her/his comments and suggestions. Fe is mainly present at +2 and +3 valences, cf. Ref. 39. Nitrogen doping into graphene exists in many different configurations, oxidized N being one of them, as well as pyridine N, pyrrole N, etc.

  • Line 154 “The Faraday efficiency (FE) at different mass ratios is closely related to the applied spot position” What is spot position? This needs to be explained.

Response: We thank the reviewer for her/his comments and suggestions. There was a problem with the formulation of this sentence, which should have been an added potential; we have changed it in the text. (Please see lines 163-164)

  • Line 177 “it can be speculated that C-N and Fe-N are the catalytic active sites of the CO2RR”. It is not clear how this conclusion could be made. More discussion is required and what are the sorption mechanism onto these active sites?

Response: We thank the reviewer for her/his comments and suggestions. We have discussed further the adsorption mechanisms of C-N and Fe-N as CO2RR active sites. (Please see lines 187-192)

  • Line 217 “that the reduction of CO2 exhibited good activity at lower potentials” Could you specify reaction rate explicitly.

Response: Current density is commonly used in electrochemistry to characterise the rate of reaction; the higher the current density, the faster the rate of reaction. In this experimental test at -1.0V vs RHE, the total current density reaches 29.1 mA/cm2; and is further specified in the text. (Please see lines 220-224)

11) SI.  "Put 10 mL of electrolyte in each cathode". This is not clear.

Response: We thank the reviewer for her/his comments and suggestions. We have completed the sentence. (Please see SI lines 62-63)

Reviewer 2 Report

The work that is entitled "Revealing the real role of etching during controlled assembly of nanocrystals applied to electrochemical reduction of CO2" describes the use of nitrogen-doped iron carbide catalysts loaded onto reduced graphene oxide for CO2 reduction. The work is interesting and I will suggest it for publication after addressing some minor points:

1) It is needed at the beginning of the Results paragraph to describe what is what from each sample.

2) Do the authors collect Raman spectra of graphene-based materials? Is there any effect on the graphene oxide upon the interaction with the iron catalysts?

3) It is needed to correct some english errors.

Author Response

The work that is entitled "Revealing the real role of etching during controlled assembly of nanocrystals applied to electrochemical reduction of CO2" describes the use of nitrogen-doped iron carbide catalysts loaded onto reduced graphene oxide for CO2 reduction. The work is interesting and I will suggest it for publication after addressing some minor points:

  • It is needed at the beginning of the Results paragraph to describe what is what from each sample.

Response: We thank the reviewer for her/his comments and suggestions. We have described each sample in the results paragraph. (Please see lines 220-224)

  • Do the authors collect Raman spectra of graphene-based materials? Is there any effect on the graphene oxide upon the interaction with the iron catalysts?

Response: We thank the reviewer for her/his comments and suggestions. We collected Raman spectra of graphene-based materials and analysed their interactions with iron catalysts. (Please see lines 132-140)

  • It is needed to correct some english errors.

Response: We thank the reviewer for her/his comments and suggestions. we have corrected the english errors in the manuscript. (Please see the revised manuscript)

Reviewer 3 Report

1-regarding the low faradaic efficiency and low current density of the prepared electrocatalyst, I would like to ask the authors what is the benefit of this electrocatalyst compared to the state of the art catalysis? 

2- regarding the synthesis of the electrocatalyst, as temprature of 800 C is required, I would like to ask the authors how they would explain the energy consumption in preparation of the catalyst and efficiency of the catalyst. 

3- . How iR is corrected is not clearly discussed in the experimental section. 

Author Response

  • regarding the low faradaic efficiency and low current density of the prepared electrocatalyst, I would like to ask the authors what is the benefit of this electrocatalyst compared to the state of the art catalysis? 

Response: As Table S1 compares recent catalysts, the catalysts we prepared require a lower potential compared to recent catalysts, which allows for lower energy efficiency.

  • regarding the synthesis of the electrocatalyst, as temprature of 800 ℃ is required, I would like to ask the authors how they would explain the energy consumption in preparation of the catalyst and efficiency of the catalyst. 

Response: High temperature calcination changes the crystalline phase of the metal (conversion of Fe3O4 NPs to Fe3C NPs); removes volatile substances and retains certain chemical components, thus giving the catalyst a stable activity; moderates the intrinsic tension of the molecular structure and improves stability.

3) How iR is corrected is not clearly discussed in the experimental section. 

Response: iR corrections were not performed during electrochemical testing in this paper and are described in the SI. (Please see SI lines 67-68)

Round 2

Reviewer 1 Report

The authors answered most of my questions, the revised version looks much better. Still I have a few additional comments based on the new information they provided:

1) What is the ion flame detector? Do you mean Flame ionization detector? (FID)

2) The description should be written in the 3rd person. For example, instead of “Place 10 mL of 0.5 M KHCO3 electrolyte in each chamber and leave a certain headspace” it should be “10 mL of a 0.5 M KHCO3 electrolyte was placed in each chamber providing a certain headspace”.

3) Supplementary information lines 8-13. Specify whether % are wt.% or vol.% ?

4) All abbreviations (CO2RR, FE) should be replaced with full names in the abstract and in the conclusions.

5) Abstract: the high selectivity of FE =40.8%. Use correct terminology as selectivity and faraday efficiency have different meaning. Selectivity accounts for all possible steps: chemical and electrochemical, while Faraday efficiency accounts only for electrochemical steps.  It would be good to provide the formulas for selectivity and FE in the supplementary data?  Also 40% cannot be seen as high. I would say moderate instead.

6) The following sentence should be reworded as the products cannot be with superior stability. This does not make sense:

 ”The main products of the N-Fe3C/rGO-H electrocatalytic reduction of carbon dioxide were CO and H2, tested in a 0.5 M KHCO3 electrolyte at room temperature and pressure, and with superior stability” .

Author Response

The authors answered most of my questions, the revised version looks much better. Still I have a few additional comments based on the new information they provided:

  • What is the ion flame detector? Do you mean Flame ionization detector? (FID)

Response: Yes, we have changed it. (Please see SI lines 67)

  • The description should be written in the 3rd person. For example, instead of “Place 10 mL of 0.5 M KHCO3 electrolyte in each chamber and leave a certain headspace” it should be “10 mL of a 0.5 M KHCO3 electrolyte was placed in each chamber providing a certain headspace”.

Response: We thank the reviewer for her/his comments and suggestions. We have used the 3rd person description in the manuscript. (Please see SI lines 62-63)

  • Supplementary information lines 8-13. Specify whether % are wt.% or vol.% ?

Response: We thank the reviewer for her/his comments and suggestions. We have specified the %, but the purity of both octadecene and oleic acid was determined using GC and calculated according to the area normalisation method. Strictly speaking it cannot be said to be a percentage by weight, but it is related to the mass, so I still use % for these two drugs to express the purity. (Please see SI lines 8-11)

4) All abbreviations (CO2RR, FE) should be replaced with full names in the abstract and in the conclusions.

Response: We thank the reviewer for her/his comments and suggestions. We have replaced the abbreviations in the abstract and conclusions with the full names. (Please see abstract and conclusions)

5) Abstract: the high selectivity of FE =40.8%. Use correct terminology as selectivity and faraday efficiency have different meaning. Selectivity accounts for all possible steps: chemical and electrochemical, while Faraday efficiency accounts only for electrochemical steps.  It would be good to provide the formulas for selectivity and FE in the supplementary data?  Also 40% cannot be seen as high. I would say moderate instead.

Response: We thank the reviewer for her/his comments and suggestions. We have changed the correct terminology and added the calculation formula to the supplementary data. (Please see SI lines 69-75)

6) The following sentence should be reworded as the products cannot be with superior stability. This does not make sense: The main products of the N-Fe3C/rGO-H electrocatalytic reduction of carbon dioxide were CO and H2, tested in a 0.5 M KHCO3 electrolyte at room temperature and pressure, and with superior stability” .

Response: We thank the reviewer for her/his comments and suggestions. We have reworded the sentence. (Please see lines 14-16)

Reviewer 3 Report

the concerns have been justified.

Author Response

We thank the reviewer for her/his comments and suggestions.